# Constructing Applied Knowledge in Nursing Students: A Learning Experience Centered on Evidence-Based Practice

**DOI:** 10.3390/nursrep15020041

**Published:** 2025-01-26

**Authors:** Judith García-Expósito, Glòria Tort-Nasarre, Alba Torné-Ruiz, Judith Roca, Sara Esqué, Montserrat Sanromà-Ortíz

**Affiliations:** 1School of Nursing, University of Andorra, De la Germandat Sq, 7, AD600 Sant Julià de Lòria, Andorra; jgarciaes@uda.ad (J.G.-E.); sesque@uda.ad (S.E.); 2Faculty of Health Sciences, Universitat Oberta de Catalunya, Rambla del Poblenou, 156, 08018 Barcelona, Spain; 3Department of Nursing and Physiotherapy, Faculty of Nursing and Physiotherapy, University of Lleida, 25198 Lleida, Spain; gloria.tort@udl.cat (G.T.-N.); montserrat.sanroma@udl.cat (M.S.-O.); 4Hospital Fundació Althaia, Xarxa Assistencial Universitària de Manresa, 08243 Manresa, Spain; 5Health Education, Nursing, Sustainability and Innovation Research Group (GREISI), IRLLeida, 25198 Lleida, Spain

**Keywords:** evidence-based practice, evidence-based nursing, students, nursing care, cooperative learning, online discussion forum

## Abstract

**Background:** Evidence-based practice must be promoted in nursing education to provide quality care. For this, teaching practices that promote its development must be considered. Aim: The aim of this study was to explore a learning experience centered on evidence-based practice through cooperative learning in an online discussion forum with nursing students. **Methods:** We employed a descriptive qualitative study with the participation of 137 nursing students. A teaching intervention based on cooperative teamwork through an online forum is proposed. The students, through a question, must manage information to provide a reasoned answer. The interactions in the forums were analyzed through qualitative content analysis. **Results:** Two themes and eight categories emerged: Critical Analysis of the Literature (Computer Literacy, Reliability of the data sources, Level of evidence according to the article design, and Relevance of the research) and Clinical Practice (Patient’s perspective, Professional experience, Quality of care, and Usefulness of EBP). **Conclusions:** The use of interactive teaching methodologies (cooperative learning and online discussion forums) facilitates the construction of knowledge applied to clinical practice. Students perceive EBP as necessary and useful for optimal care management, and the forums allow for the development of key competencies, such as autonomous learning, teamwork, and critical thinking.

## 1. Introduction

Nursing, conceived as an applied science, uses knowledge derived from healthcare practice. Within this context, the integration of the best scientific evidence into clinical situations is essential to guarantee quality nursing care and to obtain optimum results in patients [1]. An extensive review of the literature [2] underlines that evidence-based practice (EBP) improves the patient’s results, creates a positive return to investment in the healthcare system and emphasizes the responsibility of educators in the preparation of future professionals. Training must be aligned with the scientific information available that has been demonstrated to have a favorable influence at the clinical level and in the efficient management of resources [2,3].

Education is essential to promote the use of EBP [4], being an indispensable element in the training of nursing professionals [5]. Nevertheless, its integration into the teaching context is a challenge [6] and students receive training that is mainly centered on research methodologies rather than EBP [2]. While the former teaches them to gather evidence and increase the available knowledge, EBP translates the findings into applications for healthcare practice.

Therefore, separating research from clinical reality is assuming the risk of offering deficient care and not obtaining the best results. Thus, critical–reflective thinking must be fostered in students, which can lead them to ask questions, improve the search for information, and reinforce a culture of permanent self-assessment [7]. These competencies are related to the EBP stages of development [8]. Nevertheless, the teaching methods for their effective teaching are still sub-optimal [9], and the literature [10] indicates that interactive and cooperative strategies are needed. Cooperative learning is a process in which students cooperate, discuss, negotiate, and reach a consensus on problems [11].

The students work as a team and share knowledge and experiences, and all the activities have a common objective [12], thus making the students responsible for their own learning [13]. In this sense, promoting independent learning in higher education is a crucial aspect that fosters independence in students and allows them to develop the confidence necessary to exert, in their future profession, nursing that is independence, and to make professional decisions as well [14,15]. Therefore, this methodology is relevant if it is considered that, in reality, EBP is the perfect competency, as in clinical contexts, it provides the empowerment to develop teamwork and to make decisions based on evidence in a joint, reflective, and intentional manner [16].

Also, it allows for the development of general competencies, such as communication and critical thinking [17,18], which are indispensable for offering quality care and to improve clinical practice [3,19]. In this education context of cooperative learning, the use of information and communication technologies, and more specifically, of online discussion forums (ODFs), is presented as a supporting tool for interactive and cooperative learning [20] and must be considered a teaching method to create a learning experience based on EBP. Recent literature [21] details that nursing students do not have the necessary knowledge and skills to develop EBP; therefore, nursing educators need to reinforce their teaching. The integration of EBP into nursing curricula involves the creation of different pedagogical resources and strategies that can facilitate learning and teaching [22].

The main aim of the study was to explore a learning experience based on EBP through cooperative learning in an online discussion forum with nursing students. The secondary objectives were (1) to enquire about the perception of the students regarding this learning experience and (2) to determine the characteristics of the communication interactions produced during the collective construction of applied knowledge.

## 2. Materials and Methods

### 2.1. Study Design

A qualitative descriptive design based on the constructive paradigm is presented. This design is oriented towards detailing the experiences and perceptions of the participants in a specific context [23]. The Standards for Reporting Qualitative Research (SRQR) checklist was followed for its development due to its flexibility and adaptation of different qualitative research methods [24].

### 2.2. Subjects and Setting

This teaching intervention was conducted with 2nd-year students enrolled in the nursing degree at the Faculty of Nursing and Physiotherapy (FIF), University of Lleida (UdL). The nursing degree in Spain is a four-year-full-time program. A non-probabilistic, convenience sampling method was utilized, considering accessibility and the greatest representation of students [25]. The inclusion criteria were students enrolled in the Adult Nursing Care II course, independently of sociodemographic aspects (age, sex, previous education, and employment, among others), to obtain the greatest heterogeneity of students. The exclusion criteria were students who did not complete all the activities planned in the course or did not sign the informed consent form. A total of 137 participated in the study, from a total population of 146. The sample was mostly female (107 out of 137, 78.10%), with a mean age of 20.55 years old (SD = 1.64).

### 2.3. Ethical Issues and Approval

The study was positively evaluated by the FIF Study Commission, according to ethical criteria and academic honesty, and was funded by the Dean’s office as a teaching innovation and by the research studies committee (CAERFIF). The recruitment was performed by a professor who was not directly involved in the development of this teaching intervention. The confidentiality of the data and the privacy of the participants were preserved by assigning an alpha-numeric code to each document. Participants did not receive any compensation for their participation, and their participation was completely voluntary. There were no dropouts during the intervention or resignations in the transfer of data.

### 2.4. Description of the Teaching Experience

The research team was composed of 6 researchers. The activity was adapted to the 2nd-year student’s competence level so that the EBP learning was centered on (1) a systematized search for information, (2) article selection, and (3) synthesis and qualitative assessment of the quality of the information according to the source or type of information and clinical usefulness. Figure 1 details the activities developed in this learning experience.

A subject was provided, namely caring for a tracheostomy patient. A tracheostomy is a common procedure but the aspects related to the handling and management of patients are controversial, and there is little evidence on their care [26,27]. For this, the research team provided 5 questions (Q1, Q2, Q3, Q4, and Q5) related to the subject (see Table 1).

A teaching activity was designed in two stages: Stage (1) was the first face-to-face activity in the classroom (2 h), where the teachers explained the teaching activity to be developed (orally and in writing) and randomly formed the functional groups of students (3–4 students). The students collaboratively gave the first answer to the question assigned by the teacher. After the face-to-face session, the students had one more week to complete the activity through autonomous work. And Stage (2) was an online activity through ODFs to share and agree on the best answer to the assigned question.

The learning activities were planned to be resolved cooperatively through ODFs by student groups. For this, 10 online forums were opened (2 for questions) in the Virtual Campus. The participants were randomly divided into 10 groups (G) by the professors, and a question (Q) was assigned to them (Table 1).

The forums were composed of 12–14 members organized into functional groups composed of 3 to 4 students, to facilitate participation. In the forums, only the students who had been assigned to resolve the same question could participate. The forums were opened the day after the delivery of the first activity (Stage 1) and remained active for three weeks so that the different functional groups of students could agree on the answer to the question assigned by the teacher (Stage 2).

The teaching team shared specific guidelines for ODF use. The aim of the forum was described as a space to reason and reach a consensus on information. It was specified that there was no maximum or minimum number of comments, the participation had to be as a group, the contributions must be respectful and inclusive, and that it was necessary to provide details on the bibliography consulted to avoid opinions and to enhance reasoned arguments. The communication format between the students was free, allowing participants to express their ideas in an open and reflective manner within the framework established by the assigned question. The questions were developed based on the evidence-based learning model and designed to guide students in making informed clinical decisions. These questions were constructed to be clear, open-ended, and focused on key aspects of evidence-based clinical practice and were validated through a review process by the team of researchers. Lastly, a reminder was given about the role of the professor as the facilitator.

### 2.5. Data Collection

The study data were compiled through the students’ discourses in the ODFs for 3 weeks in March 2021. Data collection focused on two types of documents generated in ODF: (1) the students’ interventions within the forum and (2) the final answer to the assigned question. All of this provided a detailed record of the interactions and decisions made by the groups. The students, according to the question assigned and their groups, had to provide an agreed-upon final answer, with this being the last intervention in the forum. The teachers created an ad hoc form to develop the final answer (Table 2).

### 2.6. Data Analysis

An inductive and deductive descriptive content analysis was performed. In the inductive analysis, the meaning units were condensed and coded into categories and themes [28]. This process was carried out through various readings of the texts, establishing a pre-analysis that allowed the meaning units to be determined in order to move on to the definition of codes that were subsequently grouped into categories and finally into themes. In the deductive analysis, the communicative interaction was examined following the assessment model of construction of knowledge and learning experiences in the ODF proposed by Gunawardena et al. [29], composed of five progressive phases: (1) Sharing and comparing information (2) Exploration of dissonances and inconsistencies between ideas and concepts, (3) Negotiation of meanings and construction of knowledge, (4) Evaluation or modification of ideas (co-construction), and (5) New agreements/application of new meanings.

The interactions can be addressed according to the participants (students-professor-content, and its different possibilities) and combined with more quantitative elements (percentages or others) [29,30]. The programs Atlas-Ti v.8 and Excel v.16.16.27 were used to analyze the data.

With respect to the saturation of data, as the base size was utilized (all the informants identified as sources of information), saturation was reached by default, as no more data were available for analysis [31]. Lastly, the meaning units were identified through the forum descriptors according to the questions (FQ1, FQ2, FQ3, FQ4, and FQ5) and the group they belonged to (G1 to G10).

### 2.7. Scientific Rigor Criteria

The qualitative rigor criteria used to ensure reliability were credibility, dependability, and transferability [32]. Credibility addresses data trustworthiness, and therefore, a maximum representation and base size convenience sampling method was used for the saturation of data. During the process of analysis, the research team (two researchers, independently) followed a standardized process (similarity and differences) in the creation of the categories. Also, with respect to dependability, the process stabilized the data. And lastly, with respect to transferability, the authors present context data that help in understanding the particular perspective.

## 3. Results

The results are presented in two sections, according to the secondary objectives: (1) perception of the students about the learning experience centered on EBP, and (2) characteristics of the interactions for the collective construction of knowledge in the ODFs. For this, 20 documents were analyzed (10 forums with the students’ discourses and 10 final answers as the last entry in the forum), and a total of 537 meaning units were identified.

### 3.1. Perception of the Students About the Learning Experience

Two themes emerged from the inductive qualitative analysis (Critical analysis of the literature and Clinical practice), as well as eight categories (Digital literacy, Reliability of the data sources, Level of evidence according to the article design, Relevance of the research, Perspective of the patient, Professional experience, Quality of care, and Usefulness of EBP), which are shown in Table 3.

### 3.2. Characteristics of the Interactions for the Collective Construction of Knowledge in the ODF

Table 4 and Table 5 show the phases proposed by Gunawardena et al. [29], the number of messages in each of them as a percentage, and meaning units. The highest number of messages was observed in phase 1, after which the number of messages progressively decreased in the other phases. Forum Q2 was the only one that developed all the phases for the construction of knowledge, and on the contrary, in Forum Q3, the students only focused on Phase 1. In this forum, the participants did not have much of a discussion and only provided information on the search of information conducted and provided a descriptive answer to the question posed.

With respect to the interactions in the ODF, the student–student interaction was the most relevant dimension and included the student–content interaction (the interaction of the student with the materials). For example, some meaning units are underlined, in which the students analyze and discuss the literature:

“Regarding the scientific evidence that you have found, together with our group, we agree that…”.FQ1_G2

“On the other hand, the idea of citation No. 4 where it talks about an antimicrobial action is surprising”.FQ1_G1

The professor plays the role of facilitator (professor–student interaction). The professor only provides specific information about the activity, fosters participation, and tries to correct mistakes:

“Indicate in the message the name of the people who make up the group, as well as the bibliography”.FQ5_G10

“The due date of the activity is very close and there is little interaction in the groups, I encourage you to participate…”.FQ4_G8

The student–professor interaction details all the messages directed to the professor. It must be highlighted that none of the groups asked the professor any questions. The messages were issued when the students answered a question asked by the professor or provided some information about the activity,

“We will take it into account and try to improve…”.FQ3_G6

“We are moving forward without much difficulty and together we resolve doubts…”.FQ4_G7

Lastly, Table 6 shows all the messages per group and the percentage of professor–student, student–professor, and student–student interactions. The highest percentage in all the groups was found in the student–student interaction (M = 88.02, SD = 3.23), followed by the professor–student (M = 7.25, SD = 3.26) and the student–professor (M = 4.71, SD = 1.32) interactions. Likewise, it can be observed that Forum Q2 shows a higher total number of messages and a higher percentage of student–professor interaction.

## 4. Discussion

The study discusses the results from a learning experience centered on EBP through cooperative work and the use of ODF with 2nd-year nursing students. This teaching experience, coinciding with other studies [10,33] brings theory closer to practice and promotes EBP knowledge by fostering critical thinking and questioning the practice. Also, it follows guidelines by authors such as Connor et al. [2], establishing a process that involves the formulation of questions, bibliographical search, the critical assessment of evidence, and its implementation.

With respect to the assessment of the usefulness of EBP, the students highlighted its ability to reduce the variability in the practice of care, in line with other authors [34]; thus promoting evidence-based care. Along these lines, other authors [35] highlight that students can improve confidence in clinical decisions by applying EBP.

The skills associated with EBP are essential in the clinical context for the safety of patients and the quality of care [36]. Also, they foster effective communication between nurses and other health professionals in conditions of equality [37]. The results indicate that cooperative work contributed towards the learning of EBP, considering its three pillars (the integration of the best research evidence, clinical experience, and the circumstances and values of patients) [38].

The students participated in cooperative work using the ODG in small groups. Studies such as those by Patelarou et al. [39] and Männistö et al. [40] have detailed the importance of innovative and interactive approaches as compared to traditional ones, demonstrating that cooperative learning in digital environments improves collaborative skills, satisfaction, and motivation of students to learn. Additionally, the results underline those students showed a good level of computer literacy and a critical attitude towards it. This reinforces the idea that the adequate management of databases promotes the development of confidence in research based on evidence for their future practice [37].

With respect to the interaction in the ODF, it is worth detailing that in consonance with other studies [13,41] that the type of interaction between students is crucial in the development of cooperative learning. In agreement with the results from other studies [29,42,43] the online discussions between students tended to be limited, without reaching high levels of knowledge construction. However, the groups with a higher participation made progress towards the advanced phases of knowledge construction with new agreements (Phase 5), while those with a lower participation were kept in the initial phases, such as exclusively comparing or sharing information (Phase 1). This phenomenon could be associated with greater participation, creating greater satisfaction and commitment to online learning [44]. However, we must question the influence of other factors, such as those related to the scientific evidence available, with respect to the different questions posed in the different forums. In this sense, having more evidence could facilitate reaching agreements faster [45] and having fewer discrepancies, but this should not compromise the critical attitude of the students toward the practice or the knowledge available [7].

Among the factors that could influence reaching higher levels of knowledge in the ODF, we find the duration, the size of the group, the level of education, the use of facilitation techniques, or the intervention of professors [42]. In the present study, these factors were present in the same form and measured in all the forums, and the progress of the knowledge in each of them was different. Likewise, it was observed that the student–professor interaction was greater in Forum Q2, which reached higher levels of knowledge overall, and that the results revealed a low interaction with the professor, a finding that coincides with other studies [46]. Along this line, Kowitlawakul et al. [47] demonstrate that independently of the technologies utilized, keeping the students involved in learning is still a challenge. One of the factors that has a direct influence on learning is personal interest [48,49] and in this sense, it is important to promote involvement in learning itself, reinforcing the autonomy of the process, through teaching strategies that have been demonstrated to improve self-motivation and enthusiasm for education [50,51,52]. Thus, the learning experience presented, and according to some pedagogic aspects that are detailed by Yeung et al. [15], such as a flexible, student-centered model centered on the student based on cooperative work in small groups and the use of technology, is shown to be effective in improving the independent learning of students, and to cultivate a more effective and committed clinical performance in their future profession. The implementation of ODF and cooperative work, along with the transversal integration of EBP in the key subjects of the nursing curriculum, would be strategies that promote information management, the analysis of cases and clinical situations, and therefore, a more dynamic nursing education, centered on the students and aligned with the current demands of nursing practice.

### Limitations and Prospective

The aim of this learning experience is for 2nd-year nursing students to become initiated in the terminology and development of EBP. In this sense, the answers provided by the students to the search questions were introductory in nature, given their level of competency. This strategy can be utilized by students in the last year of the nursing degree or master’s students if the reading and critical analysis elements are broadened. It is also worth detailing that a classroom or academic teaching strategy is presented [53] that could evolve into a second phase with a clinically integrated strategy, with the direct participation of the nurses in a healthcare context [54].

Although the short duration of the study and the homogeneity of the sample are recognized, and minimizing certain confounding variables may strengthen it, it also represents a methodological limitation and the possible transfer of the findings to other contexts. Therefore, future studies with more heterogeneous samples and in other contexts are suggested to delve deeper into the phenomenon under study. And lastly, it would be positive to consider, in other projects, aspects related to the satisfaction and commitment to learning through ODF of students and to measure the impact [36,55].

## 5. Conclusions

This learning experience demonstrates how the use of interactive methodologies, such as cooperative learning and online discussion forums, facilitates the construction of applied knowledge in nursing students in an EBP framework. The students perceived EBP as necessary and useful for the optimal management of clinical care. The forums were used as spaces that allowed cooperation between students, contributing towards the development of key competencies such as independent learning, teamwork, and critical thinking. However, this communicative interaction between participants must be fostered to achieve a more elaborate construction of knowledge. This study highlights the need to continue reinforcing cooperative training environments (between peers and with guidance from teachers) and in different modalities (online, hybrid, etc.) where students must be proactive to achieve meaningful learning that allows them to develop nursing competencies.

## Figures and Tables

**Figure 1 nursrep-15-00041-f001:**
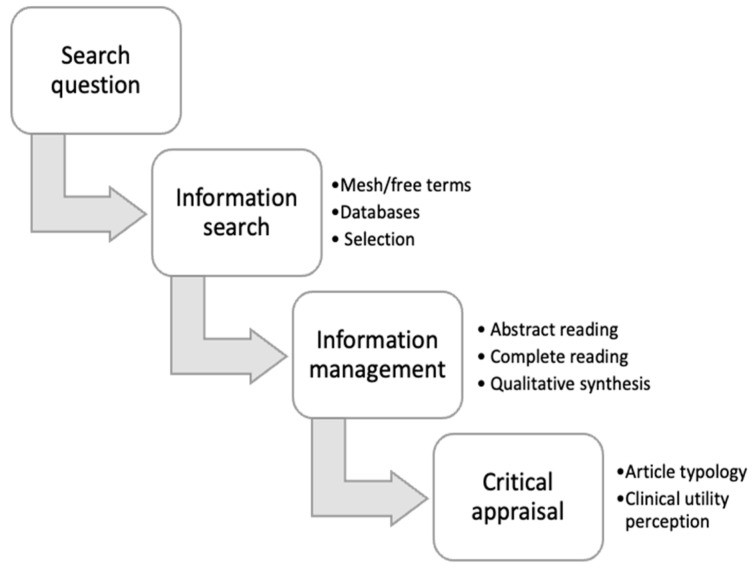
Activity development activities.

**Table 1 nursrep-15-00041-t001:** Questions and grouping of participants.

Questions	Group
^1^ Q1: When and how should decannulation be performed in a tracheotomy patient?	^2^ G1 G2
Q2: Should instillations with saline solution be made of in the bronchial tree in case of the presence of mucus in a tracheotomy patient?	G3 G4
Q3: How long before the first change, and what is the change frequency of the external cannula of the tracheotomy patient?	G5 G6
Q4: Must the change in the external and internal cannula of a tracheotomy patient be performed with a sterile technique?	G7 G8
Q5: Should the aspiration of secretions of a tracheotomy patient be performed routinely? In what situations	G9 G10

^1^ Q questions, ^2^ G group.

**Table 2 nursrep-15-00041-t002:** Final answer proposal.

Question	Activity	Agreed Upon Answer
Agreed upon answer (Q1, Q2, Q3, Q4, Q5) ^1^	Synthesis of the evidence	Summarize the evidence consulted
	Bibliographical references	Write down the references according to the Vancouver citation guidelines
	Identify the type of article you consulted ^2^	
	Meta-analysis and/or systematic reviews	
	Clinical trials	
	Quasi-experimental studies	
	Cohort or case–control studies	
	Synthesis of qualitative studies	
	Observational, analytical, and descriptive studies	
	Non-analytical studies such as case reports	
	Reports from expert committees	
	Clinical Practice Guides	
	Book chapters	
	Hospital protocols	
	Others specify…	
	Would you use the information found in real clinical practice?	(Provide an argument for the answer based on the information and type of bibliographical reference found)

^1^ Question assigned to the group, ^2^ adaptation of the Scottish Intercollegiate Guidelines Network (SIGN) model.

**Table 3 nursrep-15-00041-t003:** Themes, categories, and meaning units of the applicability of the results.

Theme	Category	Definition	Meaning Unit
Critical analysis of the literature	Digital literacy	Carry out search operations in a systematic way in databases	-Question 2: ((“Saline Solution” [Mesh] OR “Saline Solution” [tiab] OR “Crystalloid Solutions” [tiab]) AND (“Instillation, Drug” [Mesh] OR Instillation[tiab] OR Insertion[tiab] OR Infusion[tiab] OR Introduction[tiab]) AND (Bronchi[Mesh] OR Bronchi[tiab] OR Bronchus[tiab] OR Lung[tiab]) AND (Mucus[Mesh] OR Mucus[tiab])) FQ2_G3 -Question 4: ((“cannula” [MesH] OR “nasal cannula*” [MeSH] OR “nasal cannulae*” [MesH] OR “cannula*” [tiab]) AND (“Internal cannula*” [tiab]) AND (“external cannula*” [tiab]) AND “asepsis” [MeSH] OR “aseptic technique*” [tiab])) FQ4_G7
	Reliability of the data sources	The trust placed in the information	“Furthermore, it is important to keep in mind that the bibliography obtained comes from reliable sources such as the AARC (American Association for Respiratory Care) and other scientific journals mentioned above”. FQ1_G2 “…Some of the articles are more than 5 years old, and as a general rule, could be considered obsolete”. FQ1_G1
	Level of evidence according to the article design	Classification of a study according to design or type of study	“The use of PS is not recommended due to the low scientific evidence and large amount of existing biases, due to the few randomized studies on this topic”. FQ2_G3 “…These results must be taken into consideration, since they are based on a systematic review and meta-analysis of randomized controlled trials, with the purpose of evaluating the effectiveness and need of instillation of physiological saline before aspiration in ICU patients”. FQ2_G4
	Relevance of the research	Importance of research due to its ability to contribute towards knowledge development	“However, it is also important to keep in mind the importance of continuing to look after research in this area”. FQ1_G1 “For the moment, we would continue using it until we find another alternative and conduct more research that provides us with greater scientific evidence”. FQ4_G7
Clinical practice	Perspective of the patient	Understanding care from patient-centered care	“…to address the different situations of each patient individually”. FQ4_G8 “…we would give importance to the patient’s opinion and health education”. FQ4_G8
	Professional experience	Recognize experience as a modulating resource together with evidence	“…it is still carried out following the protocol...so nursing experience and knowledge is used to carry out this procedure”. FQ3_G5 “…healthcare professionals provide us with knowledge, wisdom and tools that we can use as a resource”. FQ3_G6
	Quality of care	Search for the safest and most efficient results	“The studies carried out provide maximum safety for the patient and minimize the possible errors that may arise from the procedures in order to guarantee a better quality of care or healthcare”. FQ5_G10 “So that nursing procedures are appropriate, efficient and safe”. FQ5_G9
	Usefulness of EBP	Ability to assess the benefit of its use	“Therefore, its use in clinical practice would be useful, in order to avoid variability in the performance of procedures by different health professionals”. FQ1_G2 “Therefore, it is appropriate to leave behind the routine practices that are carried out without taking into account the updated scientific evidence, as is the case of the instillation of physiological saline solution”. FQ2_G3

**Table 4 nursrep-15-00041-t004:** Percentage of knowledge construction phases.

Forum	Phase 1	Phase 2	Phase 3	Phase 4	Phase 5
Forum Q1 (G1 and G2)	91.89	6.76	1.35	-	-
Forum Q2 G3 and G4)	80.39	12.75	3.92	1.96	0.98
Forum Q3 (G5 and G6)	100	-	-	-	-
Forum Q4 (G7 and G8)	93.48	6.52	-	-	-
Forum Q5 (G9 and G10)	70.59	25.49	3.92	-	-

**Table 5 nursrep-15-00041-t005:** Meaning phased unit.

PHASE 1: Sharing and comparing information	“According to the scientific evidence found, we can corroborate what XXX has said…” FQ5_G9 “We would like to know your opinion on this and if, like us, you have found articles about this new method of bronchial instillation…” FQ2_G4 “…first of all, we found a whole set of articles, which are current, that refer to the importance of changing the patient’s cannula” FQ3_G5
PHASE 2: Exploration of dissonances and inconsistencies	“Other important measures, as pointed out by several of the functional groups and which could be different, are “…” we must continue searching…” FQ1_G2 “We do not agree and that is why we have repeated the search; a publication states that aspiration of subglottic secretions is an effective measure with little risk...”FQ5_G10 “Personally, I think that there are not enough articles related to this topic and many more searches should be done to have more scientific evidence, since we have not been able to delve deeper…” FQ2_G3 “Hello, a new question, the cannula change that is carried out after the first one should be done every 30 days or it could be done when necessary, depending on the patient’s needs. Thanks very much” FQ4_G7
PHASE 3: Negotiation of meanings and construction of knowledge	“In our group we have made a summary of the meaning of the instillations, and we could put it together to see what you think and we decide” FQ2_G3 “So, we all came to the same conclusion that the aspiration technique in tracheostomized patients is not a routine technique, right?” FQ5_G10 “It is necessary to add that focusing on the necessary health education concerning this point would reduce the incidence of morbidity and mortality of people with tracheostomy” FQ1_G2
PHASE 4: Evaluation or modification of ideas (co-construction)	“After the discussion between functional groups and the verification of the data, the following conclusions have been reached…”FQ2_G3 “We agree with the new conclusion, and with the evaluation of the selected articles, especially the ones based on systematic reviews …” FQ2_G4 “Finally, and due to the changing situation in which we find ourselves, health professionals will have to carefully assess the indications, precautions and postoperative care of patients with tracheostomies on an individual basis” FQ2_G3
PHASE 5: New agreements/application of new meanings	“… thanks to this search for evidence we will achieve greater safety in the healthcare field, both for the health personnel to perform the procedure and for the patient being treated” FQ2_G4

**Table 6 nursrep-15-00041-t006:** Messages issued by teachers and students and their interactions.

Forums	Total Messages	Professor–Student Interaction (%)	Student–Professor Interaction (%)	Student–Student Interaction (%)
Forum Q1	74	5.41	5.41	89.10
Forum Q2	102	7.84	6.86	85.30
Forum Q3	33	9.09	3.03	87.88
Forum Q4	46	2.17	4.35	93.48
Forum Q5	51	11.76	3.92	84.32
M ^1^ (SD) ^2^		7.25 (3.26)	4.71 (1.32)	88.02 (3.23)

^1^ M: Median, ^2^ SD: standard deviation.

## Data Availability

The data analyzed for the current study are not publicly available due to privacy restrictions but are available from the corresponding authors upon reasonable request.

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
