# Peer review of "Constructing Applied Knowledge in Nursing Students: A Learning Experience Centered on Evidence-Based Practice"

_nursrep, 2025, doi:10.3390/nursrep15020041_

Round 1

Reviewer 1 Report

Comments and Suggestions for Authors

Dear editor, this research is valuable. It provides important ideas. I wish you success.

1.Were participants provided feedback on their findings?

2.How many people refused to participate or dropped out of the study and why?

3.What methodological orientation is stated to support the study? For example. It should be stated that it is basic theory, discourse analysis, ethnography, phenomenology. It is stated as content analysis that this is the way the data is evaluated.

Did the authors have a guide?

Author Response

Reviewer 1. Manuscript ID: nursrep-3378484

Dear editor, this research is valuable. It provides important ideas. I wish you success.

Thanks very much for your words, and your revision.

Reviewer 1 Comment

Author Comments

Exact amendment

Were participants provided feedback on their findings?

We confirm that the students received feedback on their performance using a rubric provided by the teacher. However, this feedback was not included as part of the data analyzed in this research, as it was not part of the specific objectives of the study. We remain at your disposal for any clarification in this regard.

Without changes to the text

How many people refused to participate or dropped out of the study and why?

All the invited students agreed to participate voluntarily, and there were no dropouts during the course of the study. This information is presented in the following section:

Page 3, Line 116-19

What methodological orientation is stated to support the study? For example. It should be stated that it is basic theory, discourse analysis, ethnography, phenomenology. It is stated as content analysis that this is the way the data is evaluated.

Did the authors have a guide?

In the design part we detailed that it was a qualitative descriptive (methodological orientation) under the constructive paradigm. We believe that this is the most appropriate approach in an educational research context.

You are right, the content analysis section can be improved, thank you very much.

In the inductive analysis, a guide as such was not followed, in the deductive analysis it was based on the authors described  (Gunawardena et al., 1997; Ramírez et al., 2004).

Page 6, Line 184-87

Reviewer 2 Report

Comments and Suggestions for Authors

However, although the results of the study are detailed in the abstract, it may be useful to emphasise the practical clinical implications of the findings to reach more readers. The keywords summarise the scope of the study well, but a more prominent use of specific terms such as ‘cooperative learning’ and ‘online discussion forum’ could be suggested.

In the introduction, the importance of evidence-based practice (EBP) is well discussed and emphasised as a critical skill for students. However, the gaps in the literature and how the study aims to fill these gaps should be stated more clearly. Also, a more detailed discussion of the specific challenges students face in acquiring EBP skills could strengthen the context.

The study is based on research questions rather than direct hypotheses.

The sample details are quite clearly stated. It is stated that the study was conducted with 2nd year nursing students and 137 students participated in total. In addition, the criteria used in sample selection (accessibility and representativeness) and inclusion/exclusion criteria are clearly defined. However, although it was stated that the study had a homogenous sample (in terms of age and gender), the effect of this homogeneity on the findings was not discussed. This can be added as a methodological limitation.

In the study, student forum correspondence was used as the primary data source. However, no information was given about the evaluation of data collection tools in terms of validity and reliability. For example, it could be detailed whether the forum correspondence was structured or free-form and how the questions directed to the students were developed.

It is stated that the data of the study was collected for two weeks in March 2021. However, more specific stages of the process are not described.

Limitations are not clearly stated in the text.

The inclusion of only 137 students from a single university limits the generalisability of the findings. The fact that the study was conducted only with a group of students in Spain limits the applicability of the findings to different cultural contexts. The lack of evaluation of alternative methods other than the online forum reduces methodological diversity. As the study only covered a two-week period, the long-term impact on students' EBP learning is unknown.

The clarity of the method section is critical for the reliability of the study. Additions and elaborations in line with the suggestions mentioned above can increase the methodological soundness of the study. In particular, the completion of the deficiencies related to the data collection process, data analysis tools, and limitations will strengthen the academic contribution of the study.

The findings section is well structured in terms of categories and themes. However, including more direct quotations supporting the students' views in the tables and themes may contribute to the strengthening of the findings.

In the discussion section, the connection of the results obtained with the literature is strongly established. However, more concrete suggestions on how the findings of the study can be integrated into practical applications can be presented.

The conclusion is clearly stated, but the impact of the findings on the future professional roles of student nurses could be discussed in more detail. In addition, recommendations for future research could be made more specific.

The study provides an important contribution to develop evidence-based practice skills in nursing students and demonstrates the effectiveness of collaborative learning methods. However, shortcomings such as the short duration of the study, limited sample structure and the lack of applicability in different contexts limit the generalisability of the findings. The elimination of these deficiencies in future studies will enable a more comprehensive evaluation of the method.

Author Response

Reviewer 2. Manuscript ID: nursrep-3378484

Thanks very much for your revision and comments.

Reviewer 2 Comment

Author Comments

Exact amendment

However, although the results of the study are detailed in the abstract, it may be useful to emphasise the practical clinical implications of the findings to reach more readers. The keywords summarise the scope of the study well, but a more prominent use of specific terms such as ‘cooperative learning’ and ‘online discussion forum’ could be suggested.

We greatly appreciate your comments. We have revised and expanded the summary to more clearly emphasize the practical implications of our findings, adding the specific terms “cooperative learning” and “online discussion forum”, and so improving the visibility of the article in searches related to the topic and key focuses of the study.

Page 1, Line 25-31

In the introduction, the importance of evidence-based practice (EBP) is well discussed and emphasised as a critical skill for students. However, the gaps in the literature and how the study aims to fill these gaps should be stated more clearly. Also, a more detailed discussion of the specific challenges students face in acquiring EBP skills could strengthen the context.

We appreciate your comments on the introduction of the manuscript and have included this section to address the points made, with current bibliography.

Page 2, Line 78-83

The study is based on research questions rather than direct hypotheses.

You are absolutely right; in the end no hypotheses were included in the manuscript. This decision was made intentionally, as we consider that, in studies with a qualitative approach, hypotheses have a different role and, sometimes, can be difficult to define at the beginning of a research process. As pointed out by Amaiquema Márquez et al. (2019), research qestions allow a more open and flexible approach to the phenomenon study, which is particularly suitable for placing the reader in an exploratory and non-restrictive context. We appreciate your observation and are open to any additional suggestions in this regard.

Amaiquema Márquez et al. (2019). Approaches to formulating hypotheses in scientific research. Conrad15(70), 354-360.

Without changes to the text

The sample details are quite clearly stated. It is stated that the study was conducted with 2nd year nursing students and 137 students participated in total. In addition, the criteria used in sample selection (accessibility and representativeness) and inclusion/exclusion criteria are clearly defined. However, although it was stated that the study had a homogenous sample (in terms of age and gender), the effect of this homogeneity on the findings was not discussed. This can be added as a methodological limitation.

We recognize that homogeneity in the sample may be a limitation that must be included. For this reason, the following text has been added to the limitations section.

It should be noted that in our context, undergraduate students have a fairly homogeneous graduation profile (women with an average age of around 21 years)

Page 11, Line 342-46

In the study, student forum correspondence was used as the primary data source. However, no information was given about the evaluation of data collection tools in terms of validity and reliability. For example, it could be detailed whether the forum correspondence was structured or free-form and how the questions directed to the students were developed.

We appreciate your comment on the need to provide additional information regarding the evaluation of the data collection tool in terms of validity and reliability. We recognize that it is a crucial aspect to strengthen the rigor of the study, which is why we have expanded the corresponding section in the manuscript.

Page 4,5, Line 159-156

It is stated that the data of the study was collected for two weeks in March 2021. However, more specific stages of the process are not described.

In the section describing the teaching experience we provide greater detail about the duration of the proposed activities.

Page 4, Line 135-41

Page 4, Line 150-53

Limitations are not clearly stated in the text.

The inclusion of only 137 students from a single university limits the generalizability of the findings. The fact that the study was conducted only with a group of students in Spain limits the applicability of the findings to different cultural contexts. The lack of evaluation of alternative methods other than the online forum reduces methodological diversity. As the study only covered a two-week period, the long-term impact on students’ EBP learning is unknown.

In the limitations section we have added a text referring to your suggestions, demonstrating the limitations of the study.

It should be noted that some limitations are inherent to the nature of qualitative studies, although all approaches have benefits and limitations.

Thank you very much for your reflections.

Page 11, Line 342-46

The clarity of the method section is critical for the reliability of the study. Additions and elaborations in line with the suggestions mentioned above can increase the methodological soundness of the study. In particular, the completion of the deficiencies related to the data collection process, data analysis tools, and limitations will strengthen the academic contribution of the study.

Thank you very much for the rigor of your observations. They have allowed us to strengthen different sections and improve the work presented.

Page 4, Line 135-41

Page 4, Line 150-53

Page 4, 5, Line 159-64

Page 5, Line 170-73

Page 5, Line 175

Page 6, Line 188-91

Page 11, Line 342-46

The findings section is well structured in terms of categories and themes. However, including more direct quotations supporting the students' views in the tables and themes may contribute to the strengthening of the findings.

We appreciate your suggestion to include more detailed examples of student comments, we have created another Table.

Table 5

Page 8,9

In the discussion section, the connection of the results obtained with the literature is strongly established. However, more concrete suggestions on how the findings of the study can be integrated into practical applications can be presented.

We appreciate your comment and share the importance of giving visibility to the practical application of the results.

We have incorporated information in this regard in the final part of the discussion section.

Page 11, Line 328-32

The conclusion is clearly stated, but the impact of the findings on the future professional roles of student nurses could be discussed in more detail. In addition, recommendations for future research could be made more specific.

We take your comment into account, which is why we have added text referring to the needs detected and have included suggestions to address future work in the limitations section.

Page 12, Line 358-62

The study provides an important contribution to develop evidence-based practice skills in nursing students and demonstrates the effectiveness of collaborative learning methods. However, shortcomings such as the short duration of the study, limited sample structure and the lack of applicability in different contexts limit the generalisability of the findings. The elimination of these deficiencies in future studies will enable a more comprehensive evaluation of the method.

We appreciate your comment. In the limitations section we have added a text referring to terms of duration and sample. In addition to the possible transfer of knowledge to other contexts.

Page 11, Line 342-46

Reviewer 3 Report

Comments and Suggestions for Authors

Thank you for submitting your manuscript. This is an important topic and very relevant to nursing education.  Please see suggestions below will make this manuscript more robust.

Please expand the literature review and try to include more studies published within the past five years.  Please see example of a recent article that can be included

Estalella, I., Román, Ó., Reichenberger, T. N., & Maquibar, A. (2023). "Impact of a teaching strategy to promote evidence-based practice on nursing students’ knowledge and confidence in simulated clinical intervention choices." BMC Nursing

Impact of a teaching strategy to promote evidence-based practice on nursing students’ knowledge and confidence in simulated clinical intervention choices | BMC Nursing | Full Text 

In the Materials and Methods section please explain more clearly the process of data collection and data analysis. 

Please provide more detailed examples of student interactions in the online discussion forums (ODF).  Try to include specific instances that demonstrate critical thinking.

Author Response

Reviewer 3. Manuscript ID: nursrep-3378484

Thank you very much for your comments and suggestions of improvement.

Reviewer 3 Comment

Author Comments

Exact amendment

Please expand the literature review and try to include more studies published within the past five years.  Please see example of a recent article that can be included

Estalella, I., Román, Ó., Reichenberger, T. N., & Maquibar, A. (2023). "Impact of a teaching strategy to promote evidence-based practice on nursing students’ knowledge and confidence in simulated clinical intervention choices." BMC Nursing

You are absolutely right, and we agree that updating a quote will enrich the context of the study (Li et al., 2024; Ruzafa-Martínez et al., 2024).

We also appreciate the reference provided by Estalella et al. (0223), which we consider a valuable contribution in the context of nursing education and the promotion of evidence -based practice.

Page 2, Line 78-83

Page 10, Line 275-77

In the Materials and Methods section please explain more clearly the process of data collection and data analysis.

Thank you very much, we will explain it better. We have relocated information and detailed some elements of these sections (Methods) more explicitly.

Page 5, Line 170-73

Page 5, Line 175

Page 6, Line 188-91

Please provide more detailed examples of student interactions in the online discussion forums (ODF).  Try to include specific instances that demonstrate critical thinking.

We appreciate your suggestion to include more detailed examples of student interactions in ODFs and how they demonstrate critical thinking.

We include other examples of meaning units related to the analysis phases of the ODF, from which competency elements such as critical and reflective thinking can be inferred. 

Table 5

Page 8,9